# BiggerGait: Unlocking Gait Recognition with Layer-wise Representations from Large Vision Models

**Dingqiang Ye**[1,3]*, **Chao Fan**[2]*, **Zhanbo Huang**[3],
**Chengwen Luo**[2], **Jianqiang Li**[2], **Shiqi Yu**[1]†, **Xiaoming Liu**[3]
[1] Department of Computer Science and Engineering, Southern University of Science and Technology
[2] School of Artificial Intelligence, Shenzhen University
[3] Department of Computer Science and Engineering, Michigan State University
11810121@mail.sustech.edu.cn, chaofan996@szu.edu.cn, huang247@msu.edu,
{chengwen, lijq}@szu.edu.cn, yusq@sustech.edu.cn, liuxm@cse.msu.edu

## Abstract

Large vision models (LVM) based gait recognition has achieved impressive performance. However, existing LVM-based approaches may overemphasize gait priors while neglecting the intrinsic value of LVM itself, particularly the rich, distinct representations across its multi-layers. To adequately unlock LVM's potential, this work investigates the impact of layer-wise representations on downstream recognition tasks. Our analysis reveals that LVM's intermediate layers offer complementary properties across tasks, integrating them yields an impressive improvement even without rich well-designed gait priors. Building on this insight, we propose a simple and universal baseline for LVM-based gait recognition, termed **BiggerGait**. Comprehensive evaluations on CCPG, CAISA-B*, SUSTech1K, and CCGR_MINI validate the superiority of BiggerGait across both within- and cross-domain tasks, establishing it as a simple yet practical baseline for gait representation learning. All the models and code are available at https://github.com/ShiqiYu/OpenGait/.

## 1 Introduction

Gait recognition aims to identify an individual based on the unique patterns in the walk sequence. Unlike other biometric [23] modalities such as face [53, 8, 38, 27, 46, 45, 28, 18], fingerprint [3], or iris [51, 49], gait is unobtrusive and capable of identifying individuals from afar without their active involvement. These unique advantages make gait recognition especially effective for security applications, including suspect tracking and identity verification [47, 58, 39, 40, 29, 79, 22, 21].

To focus on pure gait patterns, early approaches suppress appearance noise by transforming each frame into pre-defined representations, like silhouettes [5, 11, 37, 57, 14], skeleton landmarks [36, 4, 63, 15], body parsing [41, 77, 80], or SMPL meshes [34, 76, 43], before feature extraction, as shown in Figure 1 (a). Although such explicit representations curb distractions from clothing and background, they also discard crucial cues: silhouettes erase body structure, skeletons remove shape information, and SMPL overly smooths personal idiosyncrasies, capping accuracy. Alternatively, recent approaches [74, 72, 26] achieve substantial gains by guiding large vision models (LVM) with human priors such as feature smoothing [74], language guidance [72], and geometry-driven denoising [26] to extract rich and implicit gait features directly from RGB data.

---

*Equal contribution. Part of the work was conducted when Mr. Ye visited MSU.
†Corresponding author.

39th Conference on Neural Information Processing Systems (NeurIPS 2025).

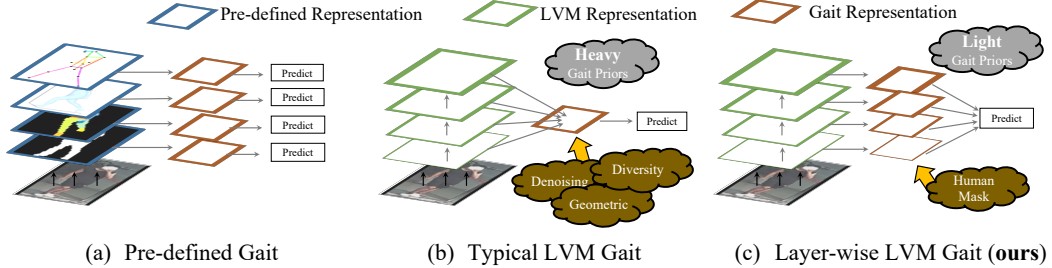

**Figure 1:** Comparison of gait representation paradigms. (a) Pre-defined gait learning uses explicit gait patterns for prediction, losing essential identity information. (b) Typical LVM gait learning relies heavily on gait priors and treats all LVM layers equally, underutilizing discriminative features. (c) Our layer-wise LVM gait learning fully utilizes intermediate LVM layers with minimal gait priors.

Despite recent advances, LVM-based gait recognition methods [74, 26] still rely heavily on traditional gait priors, whether through supervised or self-supervised training, as shown in Fig 1(b). However, is this dependence truly necessary? Through extensive experiments, we challenge this assumption, showing that LVMs [48, 30, 52] inherently possess rich, layer-wise representations that can support high-performance gait recognition with minimal reliance on human-designed priors. These findings suggest a simpler yet more robust baseline for LVM-based gait recognition, potentially redefining the role of domain knowledge in this field. Our experiments across various LVMs [52, 48, 30] and datasets [75, 56, 80, 25] further validate this remarkable observation, aligning similarly with related studies [7, 16, 59, 60, 62].

To make the above findings comprehensive, this work systematically investigates layer-wise representations in LVMs for gait recognition and uncovers three key insights: (1) Across a range of LVM architectures and scales, intermediate layers consistently yield more discriminative features than the final layer, echoing trends observed in LLMs [7, 16, 60]. (2) Each layer contributes unique, task-dependent information. (3) Intermediate-layer features are highly complementary, and their fusion produces substantial performance gains. Based on these findings, we propose a simple and universal layer-wise baseline, termed **BiggerGait**. However, fully exploiting the advantages of BiggerGait remains a challenge for GPU-limited scenes. The issue lies in the need to add a dedicated gait encoder to every LVM layer, inflating the parameters and computation cost as depth grows.

To mitigate this issue, we propose an optional grouping strategy for BiggerGait to balance performance and efficiency. Previous work [62] suggests that residual connections in LVMs encourage intermediate layers to inhabit a shared feature space. Based on this insight, we contend that a single gait encoder can effectively process features from multiple LVM layers, eliminating the need for layer-specific encoders. Specifically, this grouping strategy to merge neighboring similar LVM layers, replacing per-layer gait encoders with two shared ones: one for shallow layers and one for deep layers. This design delivers the similar performance gains as the standard BiggerGait while saving considerable cost during gait representation extraction. Experiments on CCPG [33], CCGR_MINI [80], CASIA-B* [75] and SUSTech1K [56] datasets validate the effectiveness of BiggerGait and this grouping strategy in both within- and cross-domain evaluations.

This paper makes two important contributions.

- *Comprehensive layer-wise analysis.* We conduct the first systematic examination of LVM layer representations for gait recognition, detailing how different depths affect task-specific performance and discovering that fusing complementary intermediate features unlocks substantial accuracy gains.

- *A simple yet powerful baseline.* We present BiggerGait, a simple and universal framework for LVM-based gait recognition, along with a grouping strategy to balance performance-efficiency trade-offs. Extensive experiments show that BiggerGait establishes state-of-the-art results across multiple RGB-based gait benchmarks, achieving superior performance in both intra-domain and cross-domain evaluation settings.

## 2 Related Work

**Gait Recognition** Gait recognition aims to extract subtle gait patterns that remain invariant to background clutter and clothing variations. Recent research has primarily focused on addressing these challenges in RGB video-based gait analysis [33, 56, 19, 61, 69, 66, 20]. Existing approaches can be broadly grouped into two categories: pre-defined [64, 5, 11, 37, 57, 14, 36, 63, 15, 4, 77, 41, 34, 76, 43, 71, 70, 73, 58] and LVMs representations [74, 72, 26]. Pre-defined methods explicitly extract gait-relevant components using segmentation [5, 11, 37, 57, 14], pose estimation [36, 63, 15, 4], and 3D modeling [34, 76, 43], effectively suppressing irrelevant gait information. However, this often comes at the cost of discarding identity-discriminative cues. In contrast, LVMs methods [74, 72, 26] extract implicit gait features from large vision models guided by human priors, preserving richer semantics and achieving stronger performance. Building on the LVMs paradigm, this paper introduces BiggerGait to delve deeper into its potential for gait representation.

**Large Vision Models** Motivated by the success of LLMs [2, 9, 24, 31, 54], the vision community has increasingly turned its attention to building large-scale foundation models for visual understanding [48, 30]. These models aim to learn transferable and general-purpose visual representations from massive web-scale datasets. Representative LVMs include CLIP [52], which leverages language supervision to guide visual representation learning; SAM [30], a promptable segmentation model trained on a large-scale annotated dataset for strong generalization; and DINOv2 [48], a self-supervised model that learns highly transferable features from vast, diverse image collections. The features extracted from these LVMs are termed all-purpose, as they effectively transfer to a range of downstream tasks, including image classification, semantic segmentation, and depth prediction. Our aim is to explore how these all-purpose representations can be adapted to the task of gait recognition, harnessing the broad advantages offered by LVMs. In this paper, we conduct a comprehensive investigation of how different types and scales of three representative LVMs (CLIP, SAM, and DINOv2) can be leveraged for downstream gait recognition. Further, we propose a unified baseline, BiggerGait, that consistently excels across multiple LVM architectures and model sizes.

**Layer-wise Analysis in Large Models** Recent works [60, 16, 62, 42, 17, 10, 1] have increasingly focused on layer-wise representation from large models, as intermediate features often show surprising robustness, challenging the traditional final layer representations. In NLP, researchers [42] have found that lower layers tend to encode more syntactic information, while higher layers specialize in semantic features. Others suggest that residual connections encourage layers to share a common feature space while still specializing in distinct sub-tasks [62], or that attention sink effects may weaken final-layer performance [17]. Similar trends emerge in vision domains: Head2Toe [10] selects the most useful representations from intermediate layers in transfer learning, outperforming the final layer. A fresh work [1] on LVMs again suggests that the final layer may not contain the most robust visual features, and addresses this by distilling optimal intermediate features back into the final layer. Unlike prior works [10, 1] that focus on coarse-grained vision tasks (classification, detection, and tracking), our study goes a step further by validating and advancing this insight in a significantly more demanding setting, *i.e.*, a highly fine-grained recognition task. **Beyond broadening this insight, we further reveal more interesting and unexplored findings unique to gait tasks in Sec. 3.4 & 3.5.**

## 3 Layer-wise Representation Analysis

First of all, we hypothesize that the unique layer-wise heterogeneity observed in large language models [60, 62, 16, 7, 59] (LLM) may also exist in large vision models [48, 30, 52] (LVM), potentially influencing downstream gait recognition. To verify our conjecture, we introduce a simple layer-wise gait baseline, a comprehensive experiment setting, and three key questions to systematically explore the impact of different LVM layers on gait recognition. Eventually, we further analyse the computational overhead inherent in layer-wise methods and propose a mitigation option.

### 3.1 BiggerGait: A Layer-wise LVM-based Gait Baseline

We construct a simple layer-wise gait baseline, called **BiggerGait**, illustrated in Fig. 2. Given an image $x \sim p(x)$ from a walking video, a LVM projects it into multiple intermediate feature maps $\{f_i \mid i \in \{1, 2, ..., N\}\}$ with the corresponding semantic hierarchy spanning from low to high levels.

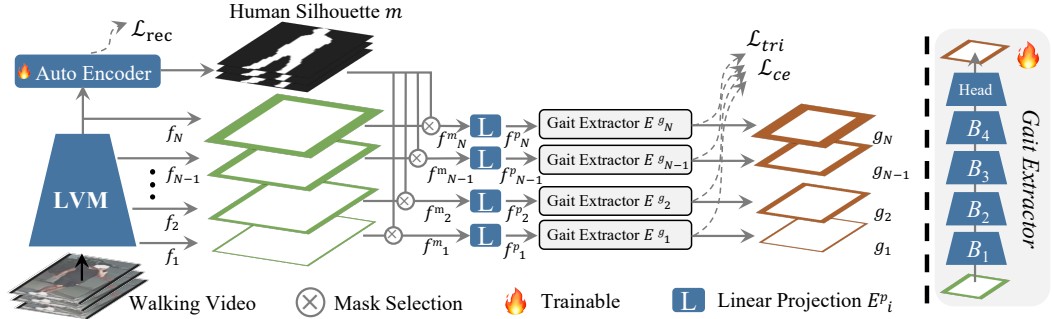

**Figure 2: Overview of the proposed BiggerGait.** An LVM extracts multi-level features from RGB videos. Human silhouettes are generated using an unsupervised auto-encoder [74], serving as the only human-designed gait prior. Each level's features, with background noise removed, are processed by separate linear projection layers and gait extractors to obtain the final gait representations.

In BiggerGait, we set $N = 12$, uniformly sampling 12 layers from the LVMs. Followed the design of BigGait [74], the last feature map $f_N$ which contains the highest-level semantic information, is fed into an auto-encoder to generate the human silhouette $m$:

$$m = \text{softmax}(E(f_N)), \ \bar{f}_N = D(m), \ \mathcal{L}_{rec} = \left\| f_N - \bar{f}_N \right\|_2^2, \tag{1}$$

where $E$ and $D$ are $1 \times 1$ convolution layers, $E$ outputs 2 channels, and $D$ restores the original channel dimension. **Notably, unlike traditional LVM gait methods [74, 26] reliant on heavy gait priors (*e.g.*, geometric, denoising, and diversity constraints), this human mask is the only gait prior we employed.** The softmax function is conducted along the channel dimension, and $\mathcal{L}_{rec}$ presents the reconstruction loss. After masking the background noise in $\{f_i\}$, we obtain $\{f_i^m\}$:

$$f_i^m = m \cdot f_i, \tag{2}$$

where $\cdot$ denote the multiplication. To reduce the GPU memory consumption, each $f_i^m$ is fed into a lightwight linear projection layer:

$$f_i^p = \text{sigmoid}(E_i^{\text{p}}(f_i^m)), \tag{3}$$

where $E_i^{\text{p}}$ consists of two $1 \times 1$ convolutions, two batch-normalization layers, a GELU activation. Its output channel is set to $C$. Here the hyper-parameter $C$ is set to 16, following [74]. Each $f_i^p$ is upsampled by bilinear interpolation to improve the resolution, *i.e.*, exhibited as a 3-D tensor with a size of $16 \times 64 \times 32$ while the first dimension denotes the output channel of the linear projection. Finally, we feed the $f_i^p$ into gait extractors $E_i^{\text{g}}$ (GaitBase [14]) to obtain gait representation $g_i$. Overall, the gait representation $R$ of the BiggerGait can be formulated as:

$$R = \{g_i = E_i^{\text{g}}(\text{sigmoid}(E_i^{\text{p}}(f_i^m))) \mid i \in \{1, 2, ..., N\}\}. \tag{4}$$

Consistent with recent works [76, 44, 67, 68, 81], triplet losses $\mathcal{L}_{tri}$ and cross-entropy losses $\mathcal{L}_{ce}$ are used for gait training. The overall loss can be formulated as:

$$\mathcal{L} = \mathcal{L}_{tri} + \mathcal{L}_{ce} + \mathcal{L}_{rec}. \tag{5}$$

In summary, the central innovation of BiggerGait lies in treating intermediate layers independently to fully unlock the power of large vision models.

## 3.2 Experimental Setting

**Separate Testing.** To assess each layer's discriminative power, we adopt a separate testing strategy at inference. Given a probe sample $x \sim \mathcal{X}$ and a gallery sample $y \sim \mathcal{Y}$, layer $i$ produces gait features $g_i^x$ and $g_i^y$. Their Euclidean distance serves as the similarity score for layer $i$:

$$d_i(x, y) = \|g_i^x - g_i^y\|_2. \tag{6}$$

Since embeddings vary across depths, each layer yields a distinct score for this pair $(x, y)$.

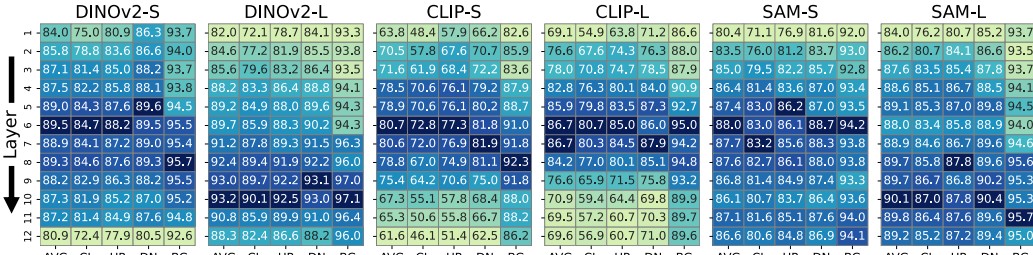

**Figure 3: Layer-wise Performance Across LVMs.** This figure presents the gait recognition accuracy of six large vision models, *i.e.*, DINOv2-Small/Large [48], CLIP-Small/Large [52], and SAM-Small/Large [30], across 12 intermediate layers, evaluated on the CCPG [33] dataset. Each cell lists the Rank-1 accuracy for full clothing (CL), top (UP), pants (DN), and bag (BG) changes, along with the average (AVG). A downward arrow on the left indicates increasing network depth. In each column, the best score is shown in black color.

**Figure 4: Layer-wise Performance Across Datasets.** The columns (A, B, C, D) represent the four test sets, CCPG [33], SUSTech1K [56], CASIA-B* [75], and CCGR_MINI [80]. All models are trained on CCPG dataset. Column A reports within-domain performance, whereas columns B–D present cross-domain results.

| Layer | DINOv2-S A | B | C | D | DINOv2-L A | B | C | D | CLIP-S A | B | C | D | CLIP-L A | B | C | D | SAM-S A | B | C | D | SAM-L A | B | C | D |
|---|---|---|---|---|---|---|---|---|---|---|---|---|---|---|---|---|---|---|---|---|---|---|---|---|
| 1 | 84.0 | 62.8 | 50.4 | 9.1 | 82.0 | 68.6 | 39.9 | 7.7 | 63.8 | 64.3 | 50.4 | 5.4 | 69.1 | 63.1 | 48.7 | 6.9 | 80.4 | 61.4 | 45.2 | 7.0 | 84.0 | 60.0 | 49.6 | 6.4 |
| 2 | 85.8 | 62.8 | 46.0 | 9.7 | 84.6 | 60.4 | 37.8 | 8.0 | 70.5 | 55.0 | 44.0 | 5.4 | 76.6 | 55.2 | 38.8 | 5.4 | 83.5 | 43.5 | 36.4 | 6.6 | 86.2 | 54.6 | 38.1 | 6.7 |
| 3 | 87.1 | 56.6 | 33.0 | 8.7 | 85.6 | 56.8 | 39.5 | 6.7 | 71.5 | 51.4 | 41.4 | 5.0 | 78.0 | 48.0 | 40.0 | 5.3 | 85.0 | 42.7 | 36.8 | 6.1 | 87.6 | 52.7 | 46.4 | 5.8 |
| 4 | 87.5 | 65.2 | 57.0 | 9.6 | 88.2 | 52.8 | 40.3 | 7.0 | 78.5 | 54.5 | 46.1 | 5.7 | 82.8 | 57.7 | 44.9 | 6.7 | 86.3 | 54.7 | 42.7 | 6.6 | 88.6 | 57.2 | 46.4 | 7.2 |
| 5 | 89.0 | 59.0 | 55.8 | 10.1 | 89.2 | 62.4 | 52.7 | 7.5 | 78.9 | 53.7 | 47.0 | 7.4 | 85.9 | 51.5 | 51.6 | 9.4 | 87.4 | 49.7 | 46.8 | 7.2 | 89.1 | 56.7 | 51.5 | 7.9 |
| 6 | 89.5 | 66.4 | 64.2 | 11.9 | 89.7 | 64.4 | 55.3 | 8.4 | 80.7 | 56.5 | 49.8 | 8.5 | 86.7 | 59.6 | 56.6 | 11.9 | 88.0 | 58.2 | 51.9 | 7.4 | 88.0 | 47.2 | 46.7 | 9.1 |
| 7 | 88.9 | 64.0 | 61.1 | 11.9 | 91.2 | 63.7 | 60.7 | 11.3 | 80.6 | 53.4 | 49.6 | 10.4 | 86.7 | 60.6 | 61.8 | 15.0 | 87.7 | 55.3 | 50.5 | 7.3 | 88.9 | 43.4 | 46.9 | 9.9 |
| 8 | 89.3 | 59.4 | 61.7 | 13.4 | 92.4 | 61.8 | 65.5 | 14.2 | 78.8 | 59.2 | 60.6 | 13.6 | 84.2 | 64.0 | 66.2 | 17.5 | 87.6 | 57.9 | 44.9 | 8.1 | 89.7 | 42.9 | 50.5 | 11.0 |
| 9 | 88.2 | 58.4 | 66.7 | 13.5 | 93.0 | 57.1 | 68.7 | 15.6 | 75.4 | 63.2 | 58.9 | 15.9 | 76.6 | 65.9 | 63.2 | 19.7 | 86.8 | 55.4 | 50.0 | 8.9 | 89.7 | 49.3 | 53.7 | 11.2 |
| 10 | 87.3 | 56.8 | 67.1 | 14.0 | 93.2 | 59.6 | 69.5 | 18.7 | 67.3 | 57.8 | 59.6 | 15.8 | 70.9 | 63.5 | 60.4 | 19.5 | 86.1 | 52.5 | 52.1 | 9.0 | 90.1 | 42.1 | 51.3 | 11.6 |
| 11 | 87.2 | 54.8 | 62.1 | 14.4 | 90.8 | 54.3 | 67.9 | 16.9 | 65.3 | 57.0 | 51.3 | 16.4 | 69.5 | 61.9 | 59.8 | 20.0 | 87.1 | 55.0 | 46.6 | 8.8 | 89.8 | 45.7 | 48.7 | 12.3 |
| 12 | 80.9 | 44.0 | 55.1 | 12.2 | 88.3 | 44.4 | 59.3 | 16.2 | 61.6 | 51.1 | 51.3 | 14.2 | 69.5 | 57.1 | 58.4 | 18.4 | 86.6 | 51.1 | 51.2 | 9.4 | 89.2 | 48.2 | 53.1 | 12.4 |

**Target LVMs.** Three representative LVMs with distinct pretraining strategies are evaluated in our experiment: SAM [30] is trained under supervised segmentation objectives, CLIP [52] follows an image-text contrastive learning approach, and DINOv2 [48] adopts self-supervised knowledge distillation. In this paper, LVM-S and LVM-L refer to the small and large versions of LVM. We test every one of the 12 layers in LVM-S, but uniformly sample 12 layers from LVM-L.

**Dataset.** Subsequent experiments are mainly conducted on four widely used clothing-variation and multi-view gait datasets: CCPG [33], CASIA-B* [75], SUSTech1K [56], and CCGR_MINI [80]. CCPG serves as the cornerstone benchmark, offering diverse full body clothing variations while masking faces and shoes to simulate real-world cloth-changing scenarios.

### 3.3 Do Middle Layers Outperform the Final Layer?

As shown in Fig. 3, to investigate whether middle layers offer stronger discriminative power than the final layer, we evaluate gait recognition accuracy across 12 layers of three popular LVMs [30, 52, 48] in different model sizes. Clearly, all LVMs achieve peak performance at middle layers rather than at the deepest one, echoing similar findings in LLMs [60, 16, 7]. For instance, in DINOv2-S [48], the highest average accuracy of 89.5% occurs at Layer 6, while the final layer drop to 80.9%.

This similar trend is also observed in both CLIP [52], which uses image-text pretraining, and SAM [30], trained with supervision, despite their different paradigms from the self-supervised DINOv2, highlighting the generality of this interesting phenomenon. Notably, even when model size increases, this middle-layer advantage remains, suggesting that deeper depth does not eliminate the representational superiority of intermediate layers. This effect is particularly evident in CLIP, where middle layers outperform both shallow and final layers by a significant margin.

> **Our answer: Yes, middle layers consistently outperform the final layer for gait recognition, regardless of LVM type and size.**

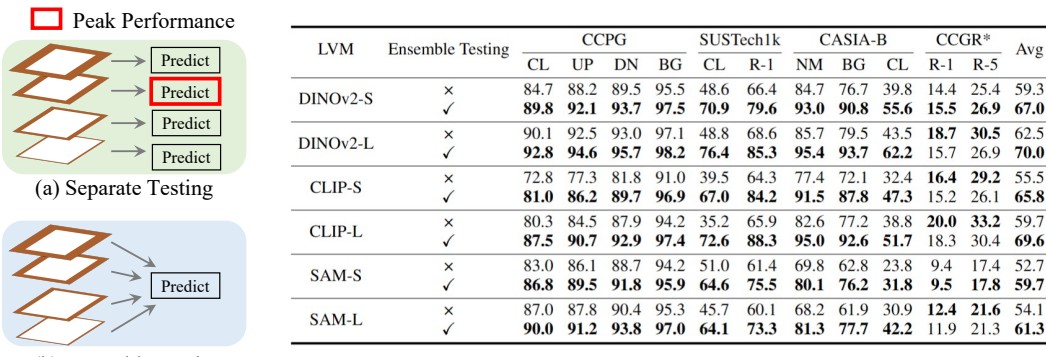

(a) Separate Testing

(b) Ensemble Testing

(c) Comparison of Separate vs. Ensemble Testing

| LVM | Ensemble Testing | CCPG | | | | SUSTech1k | | CASIA-B | | | CCGR* | | Avg |
|---|---|---|---|---|---|---|---|---|---|---|---|---|---|
| | | CL | UP | DN | BG | CL | R-1 | NM | BG | CL | R-1 | R-5 | |
| DINOv2-S | × | 84.7 | 88.2 | 89.5 | 95.5 | 48.6 | 66.4 | 84.7 | 76.7 | 39.8 | 14.4 | 25.4 | 59.3 |
| | ✓ | 89.8 | 92.1 | 93.7 | 97.5 | 70.9 | 79.6 | 93.0 | 90.8 | 55.6 | 15.5 | 26.9 | 67.0 |
| DINOv2-L | × | 90.1 | 92.5 | 93.0 | 97.1 | 48.8 | 68.6 | 85.7 | 79.5 | 43.5 | 18.7 | 30.5 | 62.5 |
| | ✓ | 92.8 | 94.6 | 95.7 | 98.2 | 76.4 | 85.3 | 95.4 | 93.7 | 62.2 | 15.7 | 26.9 | 70.0 |
| CLIP-S | × | 72.8 | 77.3 | 81.8 | 91.0 | 39.5 | 64.3 | 77.4 | 72.1 | 32.4 | 16.4 | 29.2 | 55.5 |
| | ✓ | 81.0 | 86.2 | 89.7 | 96.9 | 67.0 | 84.2 | 91.5 | 87.8 | 47.3 | 15.2 | 26.1 | 65.8 |
| CLIP-L | × | 80.3 | 84.5 | 87.9 | 94.2 | 35.2 | 65.9 | 82.6 | 77.2 | 38.8 | 20.0 | 33.2 | 59.7 |
| | ✓ | 87.5 | 90.7 | 92.9 | 97.4 | 72.6 | 88.3 | 95.0 | 92.6 | 51.7 | 18.3 | 30.4 | 69.6 |
| SAM-S | × | 83.0 | 86.1 | 88.7 | 94.2 | 51.0 | 61.4 | 69.8 | 62.8 | 23.8 | 9.4 | 17.4 | 52.7 |
| | ✓ | 86.8 | 89.5 | 91.8 | 95.9 | 64.6 | 75.5 | 80.1 | 76.2 | 31.8 | 9.5 | 17.8 | 59.7 |
| SAM-L | × | 87.0 | 87.8 | 90.4 | 95.3 | 45.7 | 60.1 | 68.2 | 61.9 | 30.9 | 12.4 | 21.6 | 54.1 |
| | ✓ | 90.0 | 91.2 | 93.8 | 97.0 | 64.1 | 73.3 | 81.3 | 77.7 | 42.2 | 11.9 | 21.3 | 61.3 |

**Figure 5: Separate vs. Ensemble Layer Testing** (a) The red box highlights the peak performance achieved by these layers. This peak performance represents the final result of separate testing. (b) Ensemble testing works likes a fair voting manner. (c) All models are trained on CCPG [33], and tested on four datasets [33, 80, 75, 56]. CCGR* indicates CCGR_MINI dataset.

## 3.4  Do Middle Layers Contribute Similarly?

To further evaluate the contribution of different LVM layers to gait recognition, we adopt additional testing datasets, each reflecting distinct characteristics. Specifically, SUSTech1K [56] emphasizes LiDAR-accessible scenes, CASIA-B* [75] represents controlled indoor environments, CCPG [33] provides extensive clothing variations, and CCGR_MINI [80] integrates diverse covariates.

We see two interesting finding in Fig. 4: (1) The best layers for cross-domain performance often differ from those for within-domain performance, occurring with a probability of 83.3%. This means that, for domain-specific tasks, the contribution of layers is inconsistent, where the most effective layer may change. (2) Notably, SUSTech1K achieves peak performance in shallow layers, most frequently at Layer 1 (66.7%), while CCGR_MINI performs better in deeper ones. We consider that deeper LVM layers capture stronger semantic features, while shallower ones preserve appearance details. To verify this, we carefully check its layer-wise performance on SUSTech1K, where the optimal layer shifts from the 1st (Same-clothing condition, easy appearance task) to the 7th (Changing-clothing condition, harder semantic task). **These results suggest that all LVM layers, from the shallowest to the deepest, may potentially benefit different domain-specific recognition tasks.**

> **Our answer: No, layer contributions vary by task, and should be treated independently.**

## 3.5  Are Middle Layers Complementary?

To evaluate the complementarity of these layers, we introduce ensemble testing during inference and compare it with separate testing, as illustrated in Fig. 5.

**Ensemble Testing.**  Unlike separate testing in Sec. 3.2, which scores each layer independently, ensemble testing pools all layer distances and yields a single unified result. For a probe-gallery pair $(x, y)$ with per-layer scores $\{d_i(x, y)\}_{i=1}^{N}$, the final similarity of ensemble testing is their mean:

$$D(x, y) = \frac{1}{N} \sum_{i=1}^{N} d_i(x, y). \qquad (7)$$

Fig. 5 (c) shows that ensemble testing often offers an impressive performance gain in both within- and cross-domain settings, raising accuracy by +7.7% on DINOv2-S, +10.3% on CLIP-S, and +7.0% on SAM-S. Meanwhile, an abnormal observation arises in the CCGR_MINI, where the fusion results sometimes slightly drop compared to the separated ones. Sec. 3.4 shows that the challenging CCGR_MINI prefers deeper semantic-based layers due to its complex data variations. We further experimented with fusing only the deeper half of the layers, which alleviate this problem on CCGR but harm generalization on other datasets such as SUSTech1K. Thus, our BiggerGait still adopts all layers for simplicity, consistency, and stronger generalization.

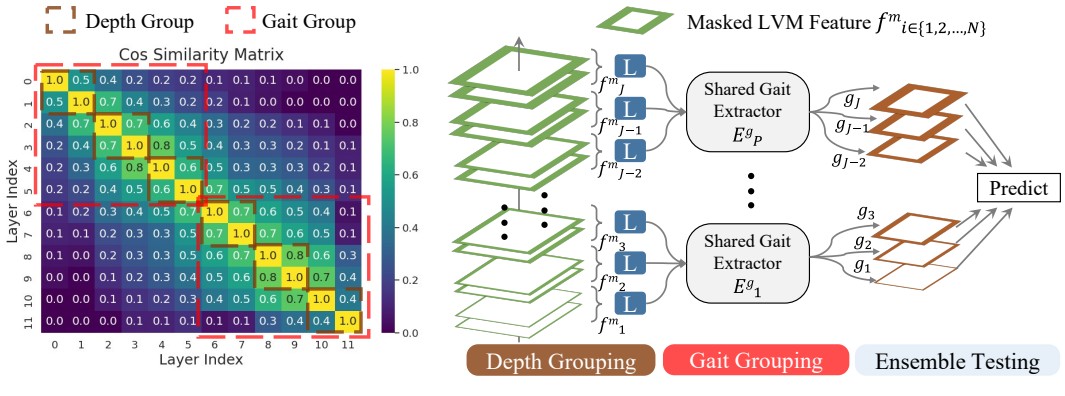

(a) Layer Similarity & Grouping Regions      (b) The architecture of BiggerGait*

**Figure 6:** (a) Average pairwise cosine similarity across the 12 layers of DINOv2-S [48], with depth and gait groups highlighted. (b) BiggerGait*: A grouped based BiggerGait that clusters similar layers ($J = 6$), uses two shared gait encoders ($P = 2$) to replace original per-layer ones, and applies ensemble testing during inference.

> **Our answer: Yes, middle layers are highly complementary, and aggregating them often yields significant gains in both within- and cross-domain tasks.**

## 3.6 Efficiency Discussion and Mitigation

Although multi-layer features are highly complementary, assigning a separate gait head to every LVM layer inflates computation and memory. Recent work [62] in LLMs shows that ViT residual connections encourage middle layers in a shared feature space. Fig. 6 (a) shows that the same pattern may also exist in LVM: for DINOv2-S [48], similarity remains strong between adjacent middle layers but decays quickly. To suit the need of GPU-limited scene, we equip BiggerGait with a grouping strategy inspired by this shared space hypothesis. To distinguish it from the standard version, we denote this grouping-based variant as **BiggerGait\***. As shown in Fig. 6 (b), the strategy has two components: (1) depth grouping, which lowers the computational cost of gait heads; and (2) gait grouping, which reduces parameter overhead.

**Depth Grouping.** Adjacent, similar layer features $\{f_i^m\}$ are divided into $J$ continuous depth groups, and the features within each group are concatenated:

$$f_j^m = \text{concat}(f_{i_1}^m, f_{i_2}^m, ..., f_{i_k}^m) \mid j \in \{1, 2, ..., J\}, \tag{8}$$

where $f_j^m$ represents the concatenated feature for group $j$. The subscript , $i_1, i_2, ..., i_k$ refer to the specific layers. Depth grouping cuts the number of gait heads' forward passes from $N$ to $J$.

**Gait Grouping.** The multiple gait encoders are also merged from $\{E_i^g\}$ into $P$ groups $\{E_p^g \mid p \in \{1, 2, ..., P\}\}$. Gait grouping reduces parameters of the gait encoders from $Ns$ to $Ps$, where $s$ is the size of one gait encoder. The gait results $R'$ of BiggerGait* is reformulated as:

$$R' = \{g_j = E_p^g(\text{sigmoid}(E_j^{\text{p}}(f_j^m))) \mid j \in \mathcal{J}, p \in \mathcal{P}\}, \tag{9}$$

where $\mathcal{J} = \{1, 2, ..., J\}$ and $\mathcal{P} = \{1, 2, ..., P\}$, denoting the number of depth groups and gait groups.

The objective of the grouping paradigm is to minimize the number of groups $J$ and $P$ while preserving the fidelity of the output representations, *i.e.*, ensuring that new gait result $R'$ closely approximates its original counterpart $R$ in Eq. (4). This objective can be expressed as a bi-level optimization problem:

$$\min_{J, P} \text{ s.t. } \min_{\{E_p^g\}, \{E_j^p\}} \|R - R'\|_2^2 \leq \epsilon. \tag{10}$$

In Sec. 4.3, we experimentally conclude that $J = 6$ and $P = 2$ represent an effective configuration. Our current grouping design prioritizes feasibility, reproducibility, and efficiency. More sophisticated grouping strategies, like explicitly maximizing representational dissimilarity or learnable grouping, are also promising future directions worth specific exploration.

**Table 1:** Performance comparison across methods and datasets. Yellow regions indicate within-domain evaluations, others are cross-domain. The last column reports the overall average accuracy. CCGR* indicates CCGR_MINI dataset. BiggerGait* refers the grouping-based BiggerGait.

| Training Dataset | Input | Method | LVM | Venue | CCPG [33] | | | | CCGR* [80] | | SUSTech1k [56] | | CASIA-B* [75] | | | Avg |
|---|---|---|---|---|---|---|---|---|---|---|---|---|---|---|---|---|
| | | | | | CL | UP | DN | BG | R-1 | R-5 | CL | R-1 | NM | BG | CL | |
| CCPG | Silh. | GaitSet [5] | - | PAMI'22 | 60.2 | 65.2 | 65.1 | 68.5 | 2.4 | 6.9 | 8.2 | 12.8 | 47.4 | 40.9 | 25.8 | 29.5 |
| | Silh. | GaitPart [11] | - | CVPR'20 | 64.3 | 67.8 | 68.6 | 71.7 | 2.4 | 6.9 | 8.1 | 13.5 | 51.2 | 41.9 | 26.0 | 30.9 |
| | Silh. | GaitGL [37] | - | ICCV'21 | 68.3 | 76.2 | 67.0 | 76.7 | 3.3 | 8.4 | 25.4 | 33.6 | 63.1 | 58.5 | 46.3 | 41.2 |
| | Silh. | GaitBase [13] | - | CVPR'23 | 71.6 | 75.0 | 76.8 | 78.6 | 2.8 | 7.3 | 9.5 | 16.8 | 59.1 | 52.7 | 30.4 | 35.6 |
| | Silh. | DeepGaitV2 [12] | - | Arxiv | 78.6 | 84.8 | 80.7 | 89.2 | 3.7 | 9.1 | 27.0 | 38.4 | 74.6 | 67.2 | 50.2 | 47.4 |
| | Silh.+Skel. | BiFusion [50] | - | MTA'23 | 62.6 | 67.6 | 66.3 | 66.0 | - | - | - | - | - | - | - | - |
| | Silh.+Skel. | SkeletonGait++ [15] | - | AAAI'24 | 79.1 | 83.9 | 81.7 | 89.9 | - | - | - | - | - | - | - | - |
| | Silh.+Pars. | XGait [78] | - | MM'24 | 72.8 | 77.0 | 79.1 | 80.5 | - | - | - | - | - | - | - | - |
| | Silh.+Pars.+Flow | MultiGait++ [25] | - | AAAI'25 | 83.9 | 89.0 | 86.0 | 91.5 | - | - | - | - | - | - | - | - |
| | RGB+Silh. | GaitEdge [35] | - | ECCV'22 | 66.9 | 74.0 | 70.6 | 77.1 | - | - | 8.9 | 19.6 | 66.5 | 58.7 | 44.8 | - |
| | RGB+Silh. | DenoisingGait [26] | SD [55] | CVPR'25 | 84.0 | 88.0 | 90.1 | 95.9 | - | - | 37.3 | 59.1 | 83.9 | 76.1 | 34.8 | - |
| | RGB | BigGait [74] | DINOv2-S | CVPR'24 | 82.6 | 85.9 | 87.1 | 93.1 | 7.4 | 16.3 | 43.7 | 56.4 | 77.4 | 71.5 | 33.6 | 53.0 |
| | RGB | BiggerGait | SAM-S [30] | Ours | 86.8 | 89.5 | 91.8 | 95.9 | 9.5 | 17.8 | 64.6 | 75.5 | 80.1 | 76.2 | 31.8 | 59.7 |
| | RGB | BiggerGait | CLIP-S [52] | Ours | 81.0 | 86.2 | 89.7 | 96.9 | 15.2 | 26.1 | 67.0 | 84.2 | 91.5 | 87.8 | 47.3 | 65.8 |
| | RGB | BiggerGait | DINOv2-S [48] | Ours | **89.8** | **92.1** | 93.7 | **97.5** | **15.5** | **26.9** | **70.9** | 79.6 | **93.0** | **90.8** | **55.6** | **67.0** |
| | RGB | BiggerGait* | SAM-S | Ours | 86.9 | 89.4 | 92.3 | 95.8 | 9.1 | 17.2 | 60.8 | 74.4 | 79.7 | 74.9 | 28.9 | 59.0 |
| | RGB | BiggerGait* | CLIP-S | Ours | 78.9 | 83.8 | 87.9 | 96.1 | 13.9 | 24.2 | 63.1 | 81.5 | 92.3 | 87.1 | 42.9 | 64.0 |
| | RGB | BiggerGait* | DINOv2-S | Ours | 89.0 | 91.9 | **94.0** | 97.2 | 14.5 | 25.3 | 69.5 | 80.4 | 91.6 | 87.7 | 54.7 | 66.5 |
| CCGR* | Silh. | GaitBase | - | CVPR'23 | 20.8 | 31.3 | **38.3** | 70.2 | 21.1 | 37.8 | 28.7 | 48.0 | 66.3 | 57.7 | 33.9 | 40.5 |
| | Silh. | DeepGaitV2 | - | Arxiv | 23.0 | 37.1 | 36.5 | 69.9 | 26.4 | 45.2 | 32.1 | 51.7 | 72.0 | 62.0 | 36.6 | 44.1 |
| | RGB | BigGait | DINOv2-S | CVPR'24 | 22.9 | 42.2 | 25.0 | 80.5 | **88.0** | **95.9** | 71.9 | 85.6 | 90.1 | 87.9 | 58.4 | 73.7 |
| | RGB | BiggerGait | SAM-S | Ours | 15.8 | 32.8 | 23.1 | 69.6 | 85.1 | 94.0 | 72.7 | 89.2 | 87.6 | 85.3 | 48.4 | 70.8 |
| | RGB | BiggerGait | CLIP-S | Ours | 21.7 | **46.5** | 28.6 | **88.4** | 80.7 | 92.0 | 81.2 | 93.3 | 94.2 | 92.8 | 60.9 | 75.7 |
| | RGB | BiggerGait | DINOv2-S | Ours | **23.2** | 44.5 | 29.5 | 86.7 | 85.7 | 94.1 | 82.1 | 93.8 | **97.2** | **96.4** | **66.7** | 78.1 |
| | RGB | BiggerGait* | SAM-S | Ours | 15.2 | 33.3 | 24.8 | 71.1 | 86.3 | 95.0 | 73.7 | 88.2 | 84.2 | 82.0 | 46.9 | 70.4 |
| | RGB | BiggerGait* | CLIP-S | Ours | 19.6 | 42.3 | 29.3 | 85.9 | 82.2 | 93.2 | 80.6 | 92.4 | 92.9 | 91.5 | 60.4 | 75.1 |
| | RGB | BiggerGait* | DINOv2-S | Ours | 22.6 | 44.9 | 31.5 | 85.8 | 87.8 | 95.8 | **83.7** | **93.8** | 96.9 | 96.3 | 65.6 | **78.5** |

# 4 Experiments

We conduct our experiments on four widely used clothing-variation and multi-view gait datasets: CCPG [33], CASIA-B* [75], SUSTech1K [56], and CCGR_MINI [80]. CCPG acts as our cornerstone benchmark, since it offers the richest wardrobe diversity, covering an array of coats, trousers, and bags in assorted color and styles, and faces and shoes are masked to emulate real-world cloth-changing scenarios. Although outfit changes are few in CCGR, it excels in covariate variety: abundant viewpoints, complex ground conditions, different walking speeds, and composite covariate scene. The full CCGR set is too large for quick testing, so we use the official mini version, which keeps the challenge but significantly drops the bulk. Therefore, the challenge CCPG and CCGR_MINI sets supply the training data, whereas evaluation is performed on all four datasets. Every experiment strictly follows the official protocols released by the owner. Gait evaluation protocols is reported for multi-view settings, and rank-1 accuracy serves as the principal metric.

## 4.1 Implementation Details

All input frames are resized to $448 \times 224$ for DINOv2 [48], $224 \times 224$ for CLIP [52] and $512 \times 256$ for SAM [30]. The training runs for 30k iterations using SGD (momentum $= 0.9$, weight-decay $= 5 \times 10^{-4}$) with an initial learning rate of $0.1$, which is dropped by 10× at 15k and 25k steps. Each mini-batch adopts the tuple $(p, k, l) = (8, 4, 30)$, which is 8 identities, 4 sequences per identity, and 30 frames per sequence. Frame sampling follows the protocol of GaitBase [13], and the sole augmentation is a random horizontal flip applied consistently to every frame within a sequence. Using eight 24GB RTX 6000 GPUs, the DINOv2-S-based BiggerGait requires approximately 8.8 hours to train on CCPG. Ensemble testing method presented in Sec. 3.5 is used during inference.

## 4.2 Main Results

To show our superiority, BiggerGait and its grouping-based variant are compared with diverse SoTA methods, including the silhouette-based [5, 11, 37, 13, 12], multimodal-based [50, 15, 35], and RGB-based [74, 35] gait methods. Due to the lack of multimodal data, cross-domain results for multimodal-based methods are unavailable.

**Table 2:** All models are evaluated on CCPG [33]. (a) & (b) Hyperparameter search for BiggerGait*. (c) Ablation study on larger LVMs. (d) Efficiency comparison across gait methods. The yellow cells in (a) & (b) mark the final setting for BiggerGait*. FLOPs are computed for an input resolution of $448 \times 224$.

(a) Ablation of Gait Group ($J = 12$)

| LVM | P | #Params | CL | UP | DN | BG |
|---|---|---|---|---|---|---|
| SAM-S [30] | 12 | 209.7M | **86.8** | **89.5** | 91.8 | 95.9 |
| | 2 | 112.2M | 85.8 | **89.5** | **91.8** | **96.3** |
| | 1 | 101.7M | 85.2 | 88.6 | 91.3 | 95.8 |
| CLIP-S [52] | 12 | 208.6M | **81.0** | **86.2** | **89.7** | **96.9** |
| | 2 | 110.8M | 80.1 | 85.5 | 89.4 | 96.3 |
| | 1 | 100.6M | 78.4 | 82.9 | 87.7 | 95.0 |
| DINOv2-S [48] | 12 | 142.2M | **89.8** | 92.1 | 93.7 | 97.5 |
| | 2 | 43.6M | 89.5 | **92.5** | **94.0** | **97.6** |
| | 1 | 33.4M | 86.3 | 90.3 | 92.5 | 97.1 |

(b) Ablation of Depth Group ($P = 2, 2, 3$)

| LVM | J | FLOPS | CL | UP | DN | BG |
|---|---|---|---|---|---|---|
| SAM-S | 12 | 95.1G | 85.8 | **89.5** | 91.8 | **96.3** |
| | 6 | 79.2G | **86.9** | 89.4 | **92.3** | 95.8 |
| | 3 | 71.2G | 84.6 | 87.5 | 90.5 | 94.4 |
| CLIP-S | 12 | 49.3G | **80.1** | **85.5** | **89.4** | **96.3** |
| | 6 | 33.4G | 78.9 | 83.8 | 87.9 | 96.1 |
| | 3 | 25.5G | 75.4 | 80.6 | 86.8 | 95.8 |
| DINOv2-S | 12 | 45.7G | **89.5** | **92.5** | **94.0** | **97.6** |
| | 6 | 29.8G | 89.0 | 91.9 | **94.0** | 97.2 |
| | 3 | 21.9G | 88.8 | 91.7 | 93.1 | 97.5 |

(c) Ablation of Scaling LVM's Size

| LVM | Method | #Params | FLOPs | CL | UP | DN | BG |
|---|---|---|---|---|---|---|---|
| SAM-L | BiggerGait | 436.4M | 258.7G | **90.0** | 91.2 | 93.8 | **97.0** |
| | BiggerGait* | 337.7M | 246.1G | 89.0 | 91.2 | 93.5 | 96.6 |
| CLIP-L | BiggerGait | 430.9M | 111.3G | **87.5** | **90.7** | 92.9 | **97.4** |
| | BiggerGait* | 332.2M | 97.0G | 85.6 | 89.7 | 91.5 | 97.2 |
| DINOv2-L | BiggerGait | 429.9M | 203.4G | **92.8** | **94.6** | 95.7 | **98.2** |
| | BiggerGait* | 339.6M | 190.7G | 92.7 | 93.9 | **96.2** | 97.8 |

(d) Parameter and GFLOPs Comparison

| Method | Upstream | Downstream | #Params | FLOPs |
|---|---|---|---|---|
| Silh.-based | DeepLabV3+ [6] | GaitBase | 34.2M | 45.4G |
| Parsing-based | SCHP [32] | GaitBase | 74.0M | 35.2G |
| Skeleton-based | HRNet-W32 [65] | Gait-TR | 29.0M | 31.2G |
| BigGait [74] | DINOv2-S | GaitBase | 30.8M | 12.7G |
| BiggerGait | SAM-S | 12 x GaitBase | 209.7M | 95.1G |
| BiggerGait | CLIP-S | 12 x GaitBase | 208.6M | 49.3G |
| BiggerGait | DINOv2-S | 12 x GaitBase | 142.2M | 45.7G |
| BiggerGait* | SAM-S | 2 x GaitBase | 112.2M | 79.2G |
| BiggerGait* | CLIP-S | 2 x GaitBase | 110.8M | 33.4G |
| BiggerGait* | DINOv2-S | 2 x GaitBase | 43.6M | 29.8G |

**Cross-domain Evaluation.** The final column of Tab. 1 highlights BiggerGait's impressive results. Trained on CCPG, the SAM-S-, CLIP-S-, and DINOv2-S-based BiggerGait impressively boost rank-1 by $+6.7\%$, $+12.8\%$, and $+14.0\%$ over prior work. With CCGR_MINI as the train set, CLIP-S- and DINOv2-S-based BiggerGait push SoTA up by $+2.0\%$ and $+4.4\%$, respectively. Such huge jumps confirm that BiggerGait learns a robust gait embedding that travels well across datasets.

Like BigGait [74], BiggerGait also shows a data-bias limitation, *i.e.*, the distribution of training data influences outcomes. Trained on CCGR_MINI and tested on CCPG, RGB-based methods exhibit less impressive results in some cases, *e.g.*, performing well in the ups-changing (UP) and bag-changing (BG), but poorly in the full-changing (CL) and pants-changing (DN). The limited clothing diversity in CCGR_MINI ($9.4\%$ of pairs, exclusively UP changes) probably accounts for this result.

**Within-domain Evaluation.** As highlighted in the yellow block of Tab. 1, BiggerGait shines on the challenge CCPG dataset. SAM-S- and DINOv2-S-based BiggerGait outperforms other methods on every metric. Notably, DINOv2-S-based BiggerGait shows significant improvements of $+5.9\%$ on CL, $+3.1\%$ on UP, $+6.6\%$ on DN, and $+4.4\%$ on BG. These results highlight BiggerGait's effectiveness in learning subtle clothing-irrelevant gait representations.

On the CCGR_MINI, BiggerGait slightly struggles. As discussed on Sec. 3.4 and 3.5, CCGR_MINI with diverse covariant only prefers deeper semantic-based layers, and other datasets prefers shallow appearance-based layers. For simplicity, consistency, and stronger generalization, BiggerGait adopts all layers, resulting slightly drop on CCGR_MINI. We consider that this performance gap is acceptable: with the same DINOv2-S backbone, BiggerGait and BiggerGait* achieves $2.3\%$ and $0.2\%$ lower rank-1 accuracy than BigGait, respectively, comparing their larger overall performance gains of $+4.4\%$ and $+4.8\%$.

**Comparing Different LVMs.** A clear domain pattern emerges: (1) the text-aligned CLIP [52] excels in cross-domain tests but lags within-domain; (2) the segmentation-supervised SAM [30] exhibits the reverse trend; (3) the self-supervised DINOv2 [48] balances both. This implies that LVM supervision strategies probably shape their domain adaptation properties thereby affecting gait recognition.

## 4.3 Ablation Study

All experiments in this section are performed on the CCPG benchmark [33]. We systematically evaluate: (1) an effective configuration for BiggerGait*; (2) the efficiency issue of BiggerGait.

**Gait & Depth Group.** Tab. 2(a) shows, using just two gait encoders delivers an accuracy similar to that of using twelve. Remarkably, even with all layer-wise features share one single gait encoder, the DINOv2-S- and SAM-S-based BiggerGait still delivers SOTA results, outperforming the methods listed in Tab. 1. Tab. 2(b) reveals that six depth groups achieve an accuracy comparable to the

**Table 3:** Model size and computation cost comparison across methods. All methods trained on CCPG, and tested on four datasets. This is a supplement for Tab. 1. This report follows the settings in Table 2 (d), with FLOPs computed at an input resolution of 448 × 224.

| Input | Method | Upstream | #Param | FLOPs | CCPG | | | | CCGR* | | SUSTech1k | | CASIA-B* | | | Avg |
|---|---|---|---|---|---|---|---|---|---|---|---|---|---|---|---|---|
| | | | | | CL | UP | DN | BG | R-1 | R-5 | CL | R-1 | NM | BG | CL | |
| Silh. | GaitSet | DeepLabV3+ | 29.4M | 50.3G | 60.2 | 65.2 | 65.1 | 68.5 | 2.4 | 6.9 | 8.2 | 12.8 | 47.4 | 40.9 | 25.8 | 29.5 |
| Silh. | GaitPart | DeepLabV3+ | 31.6M | 53.0G | 64.3 | 67.8 | 68.6 | 71.7 | 2.4 | 6.9 | 8.1 | 13.5 | 51.2 | 41.9 | 26.0 | 30.9 |
| Silh. | GaitGL | DeepLabV3+ | 30.1M | 88.7G | 68.3 | 76.2 | 67.0 | 76.7 | 3.3 | 8.4 | 25.4 | 33.6 | 63.1 | 58.5 | 46.3 | 41.2 |
| Silh. | GaitBase | DeepLabV3+ | 34.2M | 45.4G | 71.6 | 75.0 | 76.8 | 78.6 | 2.8 | 7.3 | 9.5 | 16.8 | 59.1 | 52.7 | 30.4 | 35.6 |
| Silh. | DeepGaitV2 | DeepLabV3+ | 35.2M | 93.2G | 78.6 | 84.8 | 80.7 | 89.2 | 3.7 | 9.1 | 27.0 | 38.4 | 74.6 | 67.2 | 50.2 | 47.4 |
| Silh.+Skel. | BiFusion | - | - | - | 62.6 | 67.6 | 66.3 | 66.0 | - | - | - | - | - | - | - | - |
| Silh.+Skel. | SkeletonGait++ | - | - | - | 79.1 | 83.9 | 81.7 | 89.9 | - | - | - | - | - | - | - | - |
| Silh.+Pars. | XGait | - | - | - | 72.8 | 77.0 | 79.1 | 80.5 | - | - | - | - | - | - | - | - |
| Silh.+Pars.+Flow | MultiGait++ | - | - | - | 83.9 | 89.0 | 86.0 | 91.5 | - | - | - | - | - | - | - | - |
| RGB+Silh. | GaitEdge | UNet | - | - | 66.9 | 74.0 | 70.6 | 77.1 | - | - | 8.9 | 19.6 | 66.5 | 58.7 | 44.8 | - |
| RGB+Silh. | DenoisingGait | SD & DeepLabV3+ | - | - | 84.0 | 88.0 | 90.1 | 95.9 | - | - | 37.3 | 59.1 | 83.9 | 76.1 | 34.8 | - |
| RGB | BigGait | DINOv2-S | 30.8M | 12.7G | 82.6 | 85.9 | 87.1 | 93.1 | 7.4 | 16.3 | 43.7 | 56.4 | 77.4 | 71.5 | 33.6 | 53.0 |
| RGB | BiggerGait | SAM-S | 209.7M | 95.1G | 86.8 | 89.5 | 91.8 | 95.9 | 9.5 | 17.8 | 64.6 | 75.5 | 80.1 | 76.2 | 31.8 | 59.7 |
| RGB | BiggerGait | CLIP-S | 208.6M | 49.3G | 81.0 | 86.2 | 89.7 | 96.9 | 15.2 | 26.1 | 67.0 | **84.2** | 91.5 | 87.8 | 47.3 | 65.8 |
| RGB | BiggerGait | DINOv2-S | 142.2M | 45.7G | **89.8** | **92.1** | **93.7** | **97.5** | 15.5 | 26.9 | 70.9 | 79.6 | **93.0** | **90.8** | **55.6** | **67.0** |
| RGB | BiggerGait* | SAM-S | 112.2M | 79.2G | 86.9 | 89.4 | 92.3 | 95.8 | 9.1 | 17.2 | 60.8 | 74.4 | 79.7 | 74.9 | 28.9 | 59.0 |
| RGB | BiggerGait* | CLIP-S | 110.8M | 33.4G | 78.9 | 83.8 | 87.9 | 96.1 | 13.9 | 24.2 | 63.1 | 81.5 | 92.3 | 87.1 | 42.9 | 64.0 |
| RGB | BiggerGait* | DINOv2-S | 43.6M | 29.8G | 89.0 | 91.9 | **94.0** | 97.2 | 14.5 | 25.3 | 69.5 | 80.4 | 91.6 | 87.7 | 54.7 | 66.5 |

ungrouped setup, except the CLIP-S-based one. Therefore, we set $P = 2$ and $J = 6$ for BiggerGait*, cutting roughly 108.8M and 23.8G FLOPs for DINOv2-S-based one. In this setting, DINOv2-S-based BiggerGait* achieves a 44% speedup (29.89 ms / image), approaching BigGait (21.64 ms).

**Scaling the LVM Size** Tab. 2(c) shows BiggerGait* offers marginal benefits, saving FLOPs limited while hurting performance. We consider that the upstream LVM dominates computation ($\approx 87.5\%$ for DINOv2-L), basicly making BiggerGait's overhead negligible compared to the expensive LVM's cost. Therefore, for larger LVM cases, the standard BiggerGait is recommended.

**Efficiency Comparison.** Tab. 2(d) shows that BiggerGait*, especially for DINOv2-S-based one, has similar FLOPs as the popular gait methods. This result indicates that the BiggerGait's superiority stems not from increased parameters or FLOPs, but from diverse layer-wise LVM features.

**Parameter & FLOPs Comparison.** Tab. 3 further confirms that despite its significant performance gains, the DINOv2-S-based BiggerGait* maintains FLOPs and size comparable to popular gait methods. We include the cost of DeepLabV3+ (26.8M parameters, 43.7 GFLOPs) for mask extraction in silhouette-based methods. Statistics for multimodal approaches are unavailable due to reproduction difficulty, yet they clearly incur higher computation costs from additional preprocessing models.

## 5  Conclusions

This work shifts the attention of LVM-based gait research from well-designed gait priors to the fundamental properties of LVMs itself. Our comprehensive study shows that layer-wise representations in LVM contain rich, distinct gait semantics. Without relying on elaborate gait priors, integrating these diverse layer-wise features delivers substantial gains. Building on these insights, we propose BiggerGait, a simple yet universal layer-wise LVM framework for gait recognition. We systematically analyze the inherent efficiency challenges of layer-wise methods and introduce an optional mitigation strategy. Comprehensive evaluations on CCPG, CASIA-B*, SUSTech1K and CCGR_MINI reveal BiggerGait's advance in most within- and cross-domain tasks. The work may also provide inspiration for employing the layer-wise knowledge produced by LVMs for other vision tasks.

**Limitation.** While BiggerGait sets impressive results in gait tasks, its feature extraction is mainly at the image level. The temporal feature remains underexplored in this work and deserves further study. Meanwhile, predicting the most effective layers for new, unseen datasets remains an open challenge, as the optimal layer often shifts with training data and task types.

**Acknowledgement.** This work was supported partially by National Natural Science Foundation of China (Grant 62476120, 62325307, 62422312, and 62506236) and partially by National Key R&D Program of China (Grant 2020YFA0908700 and 2023YFB4704900).

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
