# OpenReview forum: "BiggerGait: Unlocking Gait Recognition with Layer-wise Representations from Large Vision Models"
_NeurIPS.cc/2025/Conference — NeurIPS 2025 poster_

### Official Review · Reviewer_AzAN · 2025-06-20

**Clarity:** 2
**Significance:** 3
**Originality:** 3
**Rating:** 4
**Confidence:** 3

**Summary:**

The paper introduces BiggerGait, a simple yet powerful framework for gait recognition using large vision models (LVMs). It reveals that intermediate layers of LVMs contain rich and complementary gait-relevant features that outperform final layers, enabling high accuracy without heavy reliance on traditional gait priors. The authors propose a grouping strategy to mitigate computational overhead by sharing encoders across similar layers. The authors have also explore the performance of different layers of LVMs for gait recognition task. Extensive experiments across multiple benchmarks demonstrate BiggerGait’s superior within- and cross-domain performance, establishing it as a new baseline in RGB-based gait recognition.

**Questions:**

1.Can the authors provide more in-depth feature-level analysis to explain why intermediate layers of LVMs consistently outperform final layers in gait recognition? Are there any visualizations or theoretical explanations that support this phenomenon?


2. How robust are the layer-wise performance observations to variations in the architecture of the Gait Extractor? Additionally, how does the quality of the human silhouette (used for masking) affect the consistency and validity of the results?


3. The experiments suggest that earlier layers of larger models (e.g., DINOv2-L) can yield stronger performance, can the authors provide deeper analysis or hypotheses as to why larger models show this behavior?


4. In cross-domain analysis, why do deeper layers perform better on more complex datasets? Can the authors offer a clearer justification for how dataset complexity interacts with layer depth in LVMs for gait-specific semantics?


5. Can the authors elaborate on whether the proposed layer grouping strategy in BiggerGait can generalize to other vision tasks such as action recognition or pose estimation? Have any preliminary tests been conducted to verify the transferability of the observed layer-wise contributions?


6. Table 1 lacks a report of model sizes and computation costs, can the authors provide a detailed comparison of parameters and FLOPs across methods, and discuss the tradeoff between accuracy and efficiency more thoroughly?

**Ethical Concerns:**

["NO or VERY MINOR ethics concerns only"]

**Final Justification:**

My concerns are well solved by the authors during the rebuttal, and I learn towards accepting this paper.

**Limitations:**

The authors mentioned limitation of the proposed method on Line 237.

**Paper Formatting Concerns:**

No major formatting issues

**Quality:**

3

**Strengths And Weaknesses:**

Strengths:
1. It is an very interesting study to explore the contribution of different LVM layers for the gait recognition tasks, which could deliver valueable insights to the community.

2. The proposed BiggerGait model achieves good performance across various evaluation settings of the gait recognition task, which can benefit the community.

3. The proposed BiggerGait model is verified to be effective across different LVMs.

Weaknesses:

1. In Section 3.3, the authors provide experimental details to illustrate that middle layers of LVM outperform the final layer. It is an interesting observation, however, the authors are encouraged to introduce more regarding why this phenomenon happens. More insights analysis of feature level should be provided by the authors.

2. I have one concern regarding the performance of different layers of LVMS, will they be effected by different structure of the Gait Extractor? (e.g., GaitExtractor with different layers). How sensible the observation is with respects to the human silhouette quality?

3. Another interesting analysis that could be added would be the layer wise analysis towards LVMs with different sizes, e.g., Dinov2-S / Dinov2-L, where we could observe that for Dinov2 and SAM, earlier layers of larger models could provide higher gait recognition performances. The authors are encouraged to add more in-depth analysis regarding why it happens.

4. In Section 3.4, the authors provided cross-domain evaluations, where the contribution of different layers show inconsistency. For the second finding, the authors claim that "This phenomenon may stem from dataset complexity." More analysis should also be provided here. Why complex dataset benefits from deeper representations especially in terms of gait recognition? The current analysis described in the paper is very general.

5. The proposed layer grouping strategy seems can be also implemented to other computer vision tasks beyond gait recognition, e.g., action recognition, how does it work on other computer vision task? Will the observations of the contribution of different layers keep consistent?

6. In Table 1, the number of parameters of different approaches should also be reported. The authors are encouraged to conduct discussion regarding the tradeoff between accuracy and efficiency of different approaches.

---

> ### Author Rebuttal · Authors · 2025-07-30
>
> ### **\#W1/Q1: Can the authors provide more in-depth feature-level analysis to explain why intermediate layers of LVMs consistently outperform final layers in gait recognition?**
>
> Appreciate this valuable question.
>
> This interesting phenomenon has been similarly observed across various vision tasks[1], such as classification, detection, and tracking, yet its underlying mechanism remains unclear. A potential theoretical explanation provided by LLM researchers is the attention sink effect [2–4], where deeper layers exhibit a disproportionate focus on a single token.
> In contrast, intermediate layers avoid these extreme attention sinks and engage in more balanced, meaningful information processing compared to shallow or deep layers [5].
>
> Sincere thanks for your deep curiosity and thoughtful thinking on this counterintuitive phenomenon. To help first-time readers better understand this anomaly, the above viewpoint will be updated in the revised manuscript.
>
> [1] Bolya, Daniel, et al. "Perception encoder: The best visual embeddings are not at the output of the network." arXiv preprint arXiv:2504.13181 (2025).
> [2] Xiao, Guangxuan, et al. "Efficient streaming language models with attention sinks." arXiv preprint arXiv:2309.17453 (2023).
> [3] Brunner, Gino, et al. "On identifiability in transformers." arXiv preprint arXiv:1908.04211 (2019).
> [4] Gu, Xiangming, et al. "When attention sink emerges in language models: An empirical view." arXiv preprint arXiv:2410.10781 (2024).
> [5] Barbero, Federico, et al. "Why do LLMs attend to the first token?" arXiv preprint arXiv:2504.02732 (2025).
>
> ---
>
> ### **\#W2/Q2(1): How robust are the layer-wise performance observations to variations in the architecture of the Gait Extractor? (e.g., Gait Extractor with different layers)**
>
> Thanks for asking!
>
> Our Gait Extractor is a widely used GaitBase [6], adopted in many related gait works[7][8]. In our BiggerGait, we use the simplest version of GaitBase, which is based on a ResNet9 backbone. According to your suggestion, we double its depth to ResNet18 and then find that the CL performance of DINOv2-S-based BiggerGait* on CCPG drops from 89.0% to 83.7%. Although it still surpasses other SoTA methods (82.6% on BigGait), we observe that a deeper Gait Extractor tends to overfit the training data, ultimately degrading performance. This occurs because Gait Extractor's inputs are already highly refined representations from powerful LVMs, making a lightweight Gait Extractor architecture sufficient. Therefore, we consider that BiggerGait holds substantial robustness to the Gait Extractor architecture, and now the lightweight version presents a good choice.
>
> [6] Fan, Chao, et al. "Opengait: Revisiting gait recognition towards better practicality." Proceedings of the Computer Vision and Pattern Recognition Conference. 2023.
> [7] Ye, Dingqiang, et al. "Biggait: Learning gait representation you want by large vision models." Proceedings of the Computer Vision and Pattern Recognition Conference. 2024.
> [8] Jin, Dongyang, et al. "On Denoising Walking Videos for Gait Recognition." Proceedings of the Computer Vision and Pattern Recognition Conference. 2025.
>
> ---
>
> ### **\#W2/Q2(2): How does the quality of the human silhouette (used for masking) affect the consistency and validity of the results?**
>
> Performance of DINOv2-S-based BiggerGait* on CCPG under degraded human silhouette quality:
>
> - **(a) Gaussian blur (kernel=5, sigma=1.0)** introduces moderate degradation:
> **CL**: 89.0% → 87.4% (**-1.6%**)
> **UP**: 91.9% → 90.8% (**-1.1%**)
> **DN**: 94.0% → 93.5% (**-0.5%**)
> **BG**: 97.2% → 97.2% (**-0.0%**)
>
> - **(b) Additive noise (std=0.5)** results in more noticeable drops:
> **CL**: 89.0% → 83.5% (**-5.5%**)
> **UP**: 91.9% → 87.0% (**-4.9%**)
> **DN**: 94.0% → 91.3% (**-2.7%**)
> **BG**: 97.2% → 95.9% (**-1.3%**)
>
> - **(c) Dilation (kernel=3)** shows a similar degradation pattern as noise:
> **CL**: 89.0% → 82.9% (**-6.1%**)
> **UP**: 91.9% → 86.6% (**-5.3%**)
> **DN**: 94.0% → 91.4% (**-2.6%**)
> **BG**: 97.2% → 95.9% (**-1.3%**)
>
> We acknowledge that low-quality human silhouettes can impact BiggerGait’s performance to some extent. However, even under such bad conditions, BiggerGait still outperforms other SoTA methods that rely on clean and undisturbed human silhouettes (e.g., BigGait achieves 82.6% CL, 85.9% UP, 87.1% DN, and 93.1% BG as shown in Table 1), further highlighting BiggerGait’s robustness. Notably, this result reinforces our core idea: BiggerGait’s performance gains mainly come from its unique inter-layer design, rather than from the high-quality gait prior (i.e., human silhouette).
>
> ---
>
> ### **\#W3/Q3: The experiments suggest that earlier layers of larger models (e.g., DINOv2-L) can yield stronger performance.**
>
> Many thanks for your question!
> It seems there might be a small misunderstanding, as we haven't observed such a situation.
>
> Our results show similar performance patterns across different LVM sizes:
> - **(a)** In Fig. 3, intermediate layers always achieve optimal performance even in larger LVMs, with DINOv2-L and SAM-L exhibiting slightly deeper optimal layers.
> - **(b)** Fig. 4 demonstrates that performance distributions remain similar between large LVMs and their smaller counterparts. On the contrary, we observe that variations across test datasets are more significant than those provided by LVM size. Specifically, on SUSTech1K (Column B), shallower layers tend to outperform deeper ones, while on CCGR\_MINI (Column D), deeper layers perform better.
>
> We hope the above discussion helps clarify your doubt.
>
> ---
>
> ### **\#W4/Q4: Why do deeper layers perform better on more complex datasets?**
>
> Thanks for this insightful question.
>
> We consider that deeper layers in LVM exhibit stronger semantic capabilities with progressive feature transformation, while shallower layers retain more direct appearance features. We conducted an additional experiment to verify this by testing the DINOv2-S-based BiggerGait (trained on CCPG) on SUSTech1K, observing the optimal layer depth for different sub-tasks.
>
> - For the **Normal condition** (same clothing in probe/gallery, appearance-based easy task), best performance occurs at the shallowest 1st layer.
> - For the **Clothing condition** (clothing changing in probe/gallery, semantic-based hard task), the 7th layer performs best.
>
> These results align well with the observations in Lines 164-165, again revealing that hard recognition tasks prefer deeper layers than easy tasks.
>
> Your thoughtful questions on the characteristics of the different layers are truly appreciated.
> The above enlightening discussion will be added to Sec. 3.4.
>
> ---
>
> ### **\#W5/Q5: Can BiggerGait\* generalize to other vision tasks such as action recognition or pose estimation?**
>
> We appreciate this engaging question.
> Our work primarily focuses on gait recognition, and your comments highlight an important open issue that warrants further exploration. All models and code will be made publicly available, and we would be thrilled to see researchers in other vision domains exploring the applications of the findings provided by BiggerGait*.
>
> ---
>
> ### **\#W6/Q6: Table 1 lacks a report of model sizes and computation costs.**
>
> Thanks for your kind reminders.
>
> This report follows the settings in Table 2(d), with FLOPs computed at an input resolution of 448 × 224. Since silhouette-based methods require a segmentation model to extract binary masks from RGB inputs, we include the cost of DeepLabV3+, a widely used segmentation model, introducing 26.8M parameters and 43.7 GFLOPs.
>
> Due to the difficulty of reproduction, statistics for multimodal-based methods are temporarily unavailable. Although its data is difficult to obtain, there is certainly more computation cost compared to the single-modal silhouette method, as it requires extra, expensive models for more modality preprocessing.
>
> Detailed information is summarized below, and will be updated in Tab. 1.
>
> | Method         | Upstream    | #Params | FLOPs |
> |:--------------|:-----------:|:-----------:|:------:|
> | GaitSet        | DeepLabV3+  | 29.4M        | 50.3G   |
> | GaitPart       | DeepLabV3+  | 31.6M        | 53.0G   |
> | GaitGL         | DeepLabV3+  | 30.1M        | 88.7G   |
> | GaitBase       | DeepLabV3+  | 34.2M        | 45.4G   |
> | DeepGaitV2     | DeepLabV3+  | 35.2M        | 93.2G   |
> | BigGait        | DINOv2-S    | 30.8M        | 12.7G   |
> | **BiggerGait**   | SAM-S       | 209.7M       | 95.1G   |
> | **BiggerGait**   | CLIP-S      | 208.6M       | 49.3G   |
> | **BiggerGait**   | DINOv2-S    | 142.2M       | 45.7G   |
> | **BiggerGait***  | SAM-S       | 112.2M       | 79.2G   |
> | **BiggerGait***  | CLIP-S      | 110.8M       | 33.4G   |
> | **BiggerGait***  | DINOv2-S    | 43.6M        | 29.8G   |

---

> > ### Comment · Reviewer_AzAN · 2025-08-02
> > **Further Question**
> >
> > Thank you for your detailed response. I appreciate the effort of the authors during the rebuttal.
> >
> > Most of my concerns are well addressed. I have only one further question.
> >
> > Regarding Q1, the current response is too general. It would be more impactful to provide a more focused analysis specifically related to the gait recognition task. In particular, consider offering insights based on your own observations of the features learned in this work. This would help to clarify the contribution and improve the overall significance of the study.
> >
> > I will improve my score if the authors could solve this remaining concern.
> >
> > Thanks.

---

> ### Author Response · Authors · 2025-08-04
>
> ## **Rethinking Q1** ##
>
> We sincerely appreciate your affirmation and support for our responses around Q2 to Q6.
> Regarding Q1, we would like to conduct a more in-depth analysis tailored to gait recognition from two key perspectives: the feature representations and the training objectives.
>
> ---
>
> ### **(a) From the feature representation**
>
> Gait recognition is a fine-grained task that relies on both low-level appearance cues and high-level semantic understanding.  Inspired by findings in large language models (LVM) [1,4,5]—where lower layers capture syntactic patterns and higher layers focus on semantics—we assume that there should be a similar trend in large vision models (LVMs): shallow layers emphasize visual details, while deeper ones encode abstract meaning (Actually, this is also a general assumption in the paradigm of many traditional vision models [2,3]).
>
> Our experiments (also mentioned in #W4/Q4) further validate this pattern by evaluating the DINOv2-S-based BiggerGait model, trained on CCPG, on the SUSTech1K dataset across different sub-tasks:
>
> - In the **Normal** setting (same clothing in probe / gallery; appearance-focused), the shallowest 1st layer performs best.
> - In the **Clothing** setting (clothing mismatch in probe / gallery; semantics-focused), the deeper 7th layer achieves the highest accuracy.
>
> These assumption and results indicate that the optimal layer for gait recognition lying in the middle is reasonable, as it achieves a balanced trade-off between appearance and semantic information. It also implies that multi-layer aggregation is more powerful than relying solely on the final layer for the gait recognition task, since it captures diverse-level representation.
>
> ---
>
> ### **(b) From the training objectives**
>
> The training loss function of existing LVMs is not fully consistent with those of gait recognition. Specifically:
> - **CLIP**, trained with text supervision, encourages the final layer to capture coarse semantic concepts.
> - **SAM**, trained with segmentation masks, biases the model toward fine edge details.
> - **DINOv2**, as a self-supervised method, enforces consistency with its teacher model.
> - **Gait Methods**, trained with identity supervision, guide its output toward capturing fine-grained recognition features.
>
> These LVM loss functions primarily affect the final layer, making its features less suitable for identity recognition in gait tasks.
> In contrast, intermediate layers, which are less constrained by these pretraining losses, tend to generalize better and transfer more effectively to gait recognition—especially when the LVM is frozen.
>
> **We hope that the above discussion can help clarify your concerns and inspire related research in the broader vision community. If you have any further question, feel free to let us know. We are happy to reply for you.**
>
>
> ---
>
> [1] Liu, Nelson F., et al. "Linguistic knowledge and transferability of contextual representations." *arXiv preprint* arXiv:1903.08855 (2019).
>
> [2] Lin, Tsung-Yi, et al. "Feature pyramid networks for object detection." *Proceedings of the IEEE conference on computer vision and pattern recognition.* 2017.
>
> [3] Ronneberger, Olaf, Philipp Fischer, and Thomas Brox. "U-net: Convolutional networks for biomedical image segmentation." *International Conference on Medical image computing and computer-assisted intervention.* Cham: Springer international publishing, 2015.
>
> [4] Sun, Qi, et al. "Transformer layers as painters." *Proceedings of the AAAI Conference on Artificial Intelligence.* Vol. 39. No. 24. 2025.
>
> [5] Skean, Oscar, et al. "Layer by layer: Uncovering hidden representations in language models." arXiv preprint arXiv:2502.02013 (2025).

---

> > ### Comment · Reviewer_AzAN · 2025-08-04
> > **Response to the author**
> >
> > Thank you very much for your detailed response. My concerns are mostly solved and now I learn towards accepting this paper.

---

> > > ### Author Response · Authors · 2025-08-04
> > >
> > > We truly appreciate your warm support!

---

### Official Review · Reviewer_zEBF · 2025-06-22

**Clarity:** 3
**Significance:** 3
**Originality:** 3
**Rating:** 4
**Confidence:** 3

**Summary:**

This paper proposes a gait recognition method, BiggerGait, an efficient transformer-based framework using multi-layer features from large vision models (LVMs). It aggregates both intermediate and high-level visual representations through a lightweight Groupwise Head for robust performance across diverse datasets. Experiments on GREW, SUSTech1K, and CCGR show state-of-the-art or competitive results with high scalability and generalization.

**Questions:**

1) The proposed method is worse than BigGait in the evaluation using CCGR-MINI. It is still not clear why.
2) Is there any way to predict what kind of LVMs and which layers are essential for a new, unknown dataset?
3) How robust is the method to video quality degradations (e.g., blur, noise)?

**Ethical Concerns:**

["NO or VERY MINOR ethics concerns only"]

**Final Justification:**

As the authors' response addressed most of my concerns, I kept my original score.

**Limitations:**

yes.

**Paper Formatting Concerns:**

No problem.

**Quality:**

3

**Strengths And Weaknesses:**

The paper provides a comprehensive analysis of layer-wise LVM representations for gait recognition, revealing their complementary strengths. This method achieved state-of-the-art results in most datasets. The technique is clearly explained, and the evaluation experiments are well-designed.

On the other hand, the critical layer differs in each dataset, and therefore, it may be challenging to predict which layers are essential in a new task. This means generalization is still insufficient. Also, the proposed method focuses on frame-level image features, disregarding temporal dynamics, which are crucial in gait recognition.

---

> ### Author Rebuttal · Authors · 2025-07-30
>
> ### **\#Q1: The proposed method is worse than BigGait in the evaluation using CCGR\_MINI.**
>
> Appreciate this meaningful question.
>
> - **(a) From BiggerGait aspect:** Fig. 4 and Lines 164-165 show that different senses favor different layer-wise representations. CCGR\_MINI tends to prefer deeper layer representations more strongly than other domain datasets, as it poses a more challenging data distribution, with extreme viewpoint changes (e.g., overhead probe vs. frontal gallery) and 53 covariates, including complex combinations of five primary factors (Carrying, Clothing, Road, Speed, and Walking Style). For uniformity, BiggerGait employs a straightforward design that consistently uses all layers for different datasets. When ensemble testing is applied only to the deeper half of the layers on BiggerGait*, increasing rank-1 accuracy from 87.8% to 88.1%, surpassing BigGait’s 88.0%. However, such an operation caused the cross-domain rank-1 performance on SUSTech1K to drop from 93.8% to 92.8%, meanwhile disrupting BiggerGait's uniformity. Therefore, the final version of BiggerGait opted to use all layers, ensuring simplicity, consistency, and better generalization. As shown in Lines 249-251, we also believe that its benefits outweigh the drawbacks in most practical situations (+4.8% cross-domain gains versus -0.2% within-domain loss).
>
> - **(b) From BigGait aspect:** We consider that BigGait incorporates abundant gait-specific designs, such as meticulously crafted gait prior loss functions (including diversity loss and smoothness loss). While these undoubtedly enhance its within-domain performance, its generalization capability still falls short of our BiggerGait approach.
>
> Many thanks for your careful observation and valuable questions.
> The above viewpoint will be revised into the "Within-domain Evaluation" in Sec. 4.
>
> ---
>
> ### **\#Q2: Is there any way to predict what kind of LVMs and which layers are essential for a new, unknown dataset?**
>
> Many thanks for this interesting question.
>
> - **(a) For best LVMs:**  As shown in Tab. 1 and Lines 252-255, text-aligned CLIP excels in cross-domain tasks, segmentation-supervised SAM performs well within-domain tasks, while self-supervised DINOv2 delivers strong performance on both. Based on this, we recommend using text-aligned or self-supervised LVMs for new, unknown datasets.
>
> - **(b) For best layers:**  As you mentioned, predicting which layers are essential in a new, unknown task is challenging. This remains an open question beyond our current scope, calling for future research. Actually, our further analysis shows that even on the same known test dataset, the most effective layer can shift when the training data changes. Specifically, the 6th layer of DINOv2-S-based BiggerGait achieves peak performance on SUSTech1K when trained on CCPG, but the optimal layer shifts to the 10th when trained on CCGR\_MINI. Fortunately, even though the optimal layer is dynamic, we also observe that easier recognition tasks consistently utilize shallower layers than more challenging ones. Specifically, on the SUSTech1K test set, the normal setting (no clothing change, easier task) achieves peak performance on the 1st and 7th layers when trained on CCPG and CCGR\_MINI, respectively, while the clothing setting (with clothing changes, harder task) reaches its best performance on deeper layers, i.e., 7th for CCPG and 11th for CCGR\_MINI.
>
> Appreciate your insightful comments again.
> We will append the above discussion to the Limitation section.
>
> ---
>
> ### **\#Q3: How robust is the method to video quality degradations (e.g., blur, noise)?**
>
> Performance of DINOv2-S-based BiggerGait* on CCPG under different perturbations:
>
> - **(a) Gaussian blur** (kernel=5, sigma=1.0) causes notable performance drops:
> **CL**: 89.0% → 84.3% (**-4.7%**)
> **UP**: 91.9% → 88.8% (**-3.1%**)
> **DN**: 94.0% → 91.4% (**-2.6%**)
> **BG**: 97.2% → 97.0% (**-0.2%**)
>
> - **(b) Additive noise** (std=0.1) leads to relatively milder degradation:
> **CL**: 89.0% → 84.7% (**-4.3%**)
> **UP**: 91.9% → 89.4% (**-2.5%**)
> **DN**: 94.0% → 91.4% (**-2.6%**)
> **BG**: 97.2% → 96.9% (**-0.3%**)
>
> As mentioned in line 222, BiggerGait's training data augmentation only includes flipping, so this performance drop is reasonable. Remarkably, even when degraded by perturbations, BiggerGait still outperforms other state-of-the-art methods using clear and undisturbed inputs (e.g., BigGait's 82.6% CL, 85.9% UP, 87.1% DN, and 93.1% BG on Tab. 1), further demonstrating BiggerGait's robustness and superior capability. These results align with the core idea of our paper: by fully leveraging the capabilities of large vision models, excellent performance can still be achieved even relying on fewer gait priors.

---

> > ### Comment · Reviewer_zEBF · 2025-08-04
> >
> > The authors' response addressed most of my concerns. I keep my original rating.

---

> > > ### Author Response · Authors · 2025-08-04
> > >
> > > We sincerely appreciate your positive reply and kind support!

---

### Official Review · Reviewer_jAUv · 2025-06-26

**Clarity:** 3
**Significance:** 4
**Originality:** 4
**Rating:** 5
**Confidence:** 5

**Summary:**

This paper presents a comprehensive investigation into the layer-wise representations of large Visual models (LVMs) for gait recognition. Building on these findings, the authors propose BiggerGait, a simple yet effective and universal layer-wise baseline for gait recognition. The paper makes two major contributions: (1) it offers the first in-depth, systematic analysis of LVM layer representations in the context of gait recognition, revealing how feature depth influences task performance and how combining intermediate features leads to accuracy improvements; and (2) it introduces BiggerGait, a practical and high-performing framework that achieves state-of-the-art results across multiple RGB-based gait recognition benchmarks, excelling in both intra-domain and cross-domain scenarios while offering a balanced performance-efficiency trade-off through a proposed grouping strategy.

**Questions:**

None

**Ethical Concerns:**

["NO or VERY MINOR ethics concerns only"]

**Limitations:**

Yes

**Quality:**

4

**Strengths And Weaknesses:**

Strengths
-The paper presents meaningful insights regarding intermediate-layer features in large Vision Models (LVMs), particularly highlighting that these layers yield more discriminative and complementary features for gait recognition than final layers. These findings are in line with trends in other domains (e.g., large language models), which gives the study credibility and relevance.
-The introduction of BiggerGait, a simple yet effective layer-wise baseline, contributes a practical advancement to RGB-based gait recognition systems. The systematic analysis of layer-wise feature contributions appears to be a first in the domain, adding to the paper's value.
-The paper claims extensive experiments across multiple benchmarks and evaluation settings (intra- and cross-domain), which strengthens the reliability and generalizability of its results.
Weaknesses
There is insufficient discussion of prior work on layer-wise analysis in other domains. A section reviewing how such analysis has been applied in areas like NLP or vision (e.g., CNNs, Transformers) would make the paper’s positioning clearer and more compelling.
The use of “video” versus “image” to describe inputs is inconsistent, particularly in Figure 2 versus the text. This inconsistency could confuse readers about the nature of the data used and should be resolved for clarity.
The paper’s structure could be improved. Presenting key experimental results (e.g., Section 3.3) before formally introducing the experimental setup (Section 4) breaks the logical flow and could hinder comprehension for readers unfamiliar with the dataset or methodology.

---

> ### Author Rebuttal · Authors · 2025-07-30
>
> ### **\#W1: Insufficient discussion of prior work on layer-wise analysis in other domains.**
>
> Appreciate your warm and helpful reminders.
> The following discussion will be updated into the "Layer-wise Analysis in Large Models" of Sec. 2.
> ***
> Recent works[1-7] have increasingly focused on layer-wise representation from large models, as intermediate features often show surprising robustness, challenging the traditional final layer representations.
>
> In NLP, researchers[4] have found that lower layers tend to encode more syntactic information, while higher layers specialize in semantic features. Others suggest that residual connections encourage layers to share a common feature space while still specializing in distinct sub-tasks[3], or that attention sink effects may weaken final-layer performance[5].
>
> Similar trends emerge in vision domains: Head2Toe[6] selects the most useful representations from intermediate layers in transfer learning, outperforming the final layer. A fresh work[7] on LVMs again suggests that the final layer may not contain the most robust visual features, and addresses this by distilling optimal intermediate features back into the final layer to boost performance.
>
> Unlike prior works[6,7] that focus primarily on coarse-grained vision tasks (classification, detection, and tracking), our study goes a step further by validating and advancing this insight in a significantly more demanding setting, i.e., a highly fine-grained recognition task. Beyond broadening this general insight, we further reveal more unexplored findings unique to gait recognition, as shown in Sec. 3.4 \& 3.5.
>
> [1] Skean, Oscar, et al. "Layer by layer: Uncovering hidden representations in language models." arXiv preprint arXiv:2502.02013 (2025).
> [2] Fan, Siqi, et al. "Not all layers of llms are necessary during inference." arXiv preprint arXiv:2403.02181 (2024).
> [3] Sun, Qi, et al. "Transformer layers as painters." Proceedings of the AAAI Conference on Artificial Intelligence. Vol. 39. No. 24. 2025.
> [4] Liu, Nelson F., et al. "Linguistic knowledge and transferability of contextual representations." arXiv preprint arXiv:1903.08855 (2019).
> [5] Gu, Xiangming, et al. "When attention sink emerges in language models: An empirical view." arXiv preprint arXiv:2410.10781 (2024).
> [6] Evci, Utku, et al. "Head2toe: Utilizing intermediate representations for better transfer learning." International Conference on Machine Learning. PMLR, 2022.
> [7] Bolya, Daniel, et al. "Perception encoder: The best visual embeddings are not at the output of the network." arXiv preprint arXiv:2504.13181 (2025).
>
> ---
>
> ### **\#W2: The use of "video" versus "image" to describe inputs is inconsistent.**
>
> Appreciate you pointing out this confusion.
> RGB-based gait recognition uses video data as input.
> All ambiguous uses of "image" will be revised to "video," and Fig. 2 will be updated to better reflect this.
>
> ---
>
> ### **\#W3: The paper’s structure could be improved.**
>
> Appreciate this valuable suggestion.
> To enhance readability and logical coherence, dataset descriptions and implementation details from Sec. 4 will be previewed in the experiment setting in Sec. 3.2.

---

### Official Review · Reviewer_r6GY · 2025-07-03

**Clarity:** 3
**Significance:** 3
**Originality:** 2
**Rating:** 4
**Confidence:** 4

**Summary:**

This paper presents BiggerGait, a gait recognition framework that exploits intermediate representations from LVMs. The core observation lies in that middle layers yield stronger and richer representations than final layers and propose fusing them to enhance recognition accuracy without relying on handcrafted gait priors. The method approaches gait recognition from a representation learning perspective instead of designing gait-specific priors, aiming to unlock the discriminative capacity of LVMs. Experiments on four datasets show state-of-the-art performance in both within- and cross-domain settings.

**Questions:**

1. Clarify technical novelty beyond prior representation works. The authors should clarify what new insight or mechanism your work contributes beyond simply applying known findings to gait recognition.
2. Strengthen Justification for Grouping Strategy Design. Performing an ablation comparing similarity-based grouping vs. diversity-based grouping (e.g., maximizing representational dissimilarity), or explore learnable grouping strategies.
3. Inference time. Considering including actual inference time.
4. Clarify which parts of the architecture are original versus reused from prior work, and provide a step-by-step comparison with related baselines to make the contribution more explicit.

**Ethical Concerns:**

["NO or VERY MINOR ethics concerns only"]

**Final Justification:**

Thanks for the detailed response, I would raise the rating to 4.

**Limitations:**

yes

**Paper Formatting Concerns:**

None.

**Quality:**

2

**Strengths And Weaknesses:**

Strengths
1. This paper takes a perspective by leveraging intermediate representations from large vision models instead of relying on handcrafted gait priors.
2. Comprehensive analysis of layer-wise representations across three LVMs and four datasets. The visualizations are particularly effective, such as Figure 3-6.
3. Well written and easy to understand.

Weaknesses
1. Technical novelty. The core idea of fully exploiting intermediate layers representation (Line 40, Sec 3.3) in gait recognition is somehow limited. The insight is well established in existing visual representation learning literature. For example, DINOv2 shows that intermediate layers outperform final layers on a range of vision tasks. And also, ref[1][2] suggest that intermediate layers can encode even richer and more transferable representations.
  [1] Evci, Utku, Vincent Dumoulin, Hugo Larochelle, and Michael C. Mozer. Head2toe: Utilizing intermediate representations for better transfer learning. In International Conference on Machine Learning, 2022: 6009-6033.
  [2] Raghu, Maithra, Thomas Unterthiner, Simon Kornblith, Chiyuan Zhang, and Alexey Dosovitskiy. Do vision transformers see like convolutional neural networks?. Advances in neural information processing systems, 2021: 12116-12128.
2. Architecture design. The proposed framework largely builds upon and modestly extends existing works BigGait.
3. Grouping strategy. The introduced “grouping strategy” is heuristic: it relies on pairwise cosine similarity between layers to merge the most similar layers. However, this contradicts the authors’ own claim in Sec 3.5 that middle layers are complementary. Obviously, this strategy is suboptimal. No ablation is provided to justify whether similarity-based grouping is optimal, leaving the design choice weakly supported.
4. Efficiency Analysis. Lack of inference time comparisons in Sec 3.6. The model parameters and flops are insufficient to prove efficiency.

---

> ### Author Rebuttal · Authors · 2025-07-30
>
> ### **\#W1/Q1: Clarify New Insights and Technical Novelty.**
>
> Thanks for your comments. We would like to clarify our contribution in three aspects:
>
> - **(a)** Our BiggerGait works on gait recognition, a fine-grained video-based vision task, but most related works [1,2] are for coarse-grained and image-based tasks. And we argue that adapting a promising insight to the highly fine-grained and challenging task of gait recognition substantially enriches both the insight and the task itself. In particular, as shown in Lines 143–155, we demonstrate that for the fine-grained problem of gait recognition, intermediate-layer features consistently outperform final-layer features. As noted in Lines 36–37, while this trend aligns with related works [6, 14, 43, 44, 46]—including the references [1,2] you provided, which primarily investigate coarse-grained classification tasks—our study goes a step further by validating and advancing this insight in a significantly more demanding setting. This not only reinforces the generality of the idea but also establishes its previously unexplored practical value for fine-grained gait recognition.
>
> - **(b)** Beyond broadening this general insight, we further reveal two novel findings unique to gait recognition. In Lines 157-170, we observe that the contributions of intermediate layers vary significantly across different gait recognition scenarios (i.e., datasets). Specifically, as shown in Fig. 4 and Lines 164-165, on the SUSTech1K dataset, shallow layers often outperform deeper ones. For example, the first layer of DINOv2-L achieves 68.6% accuracy, significantly surpassing the 44.4% of the final layer. Interestingly, this trend reverses on the CCGR\_MINI dataset, where the final layer of DINOv2-L achieves 16.2%, compared to just 7.7% for the first. To our knowledge, this scenario-dependent reversal has rarely been reported in the visual representation learning literature, highlighting a new perspective on the role of intermediate layers in domain-specific recognition tasks.
>
> - **(c)** Based on this second finding, in Lines 171-181, we further reveal that these intermediate layers are highly complementary across most recognition scenarios. Aggregating them often yields significant gains in both within- and cross-domain scenarios. As shown in Fig. 5, the plain ensemble testing strategy significantly boosts the average Rank-1 accuracy of four testing datasets: DINOv2-S improves by 7.7%, CLIP-S by 10.3%, and SAM-S by 7.0%. On SUSTech1K specifically, these gains are even more substantial (e.g., +13.2%, +19.9%, and +14.1%, respectively). This demonstrates not only the practical benefits of exploiting intermediate layers in gait recognition but also a new and effective methodology for enhancing generalization in fine-grained recognition tasks.
>
> Taken together, these contributions indicate that our work goes beyond reiterating the general usefulness of intermediate layers by systematically validating and extending this insight in the fine-grained and highly challenging domain of gait recognition. By revealing scenario-dependent reversals, highlighting their cross-domain complementarity, and introducing a practical layer grouping strategy, we provide new evidence and perspectives that enrich the understanding and application of large vision models for gait recognition.
>
> [1] Evci, Utku, et al. "Head2toe: Utilizing intermediate representations for better transfer learning." International Conference on Machine Learning. PMLR, 2022.
>
> [2] Raghu, Maithra, et al. "Do vision transformers see like convolutional neural networks?." Advances in neural information processing systems 34 (2021): 12116-12128.
>
> ---
>
> ### **\#W2/Q4: Architecture Difference between BigGait v.s. BiggerGait.**
>
> We believe that Fig. 1 may have led to some misunderstandings.
>
> BigGait rely on heavy gait priors but BiggerGait does not. In fact, the heavy gait priors introduced by BigGait (Fig. 1 (b)) rely on many carefully designed modules, including the mask branch, appearance branch, and denoising branch (with smoothing and diversity constraints), as well as specific strategies to combine them [3].
>
> In contrast, BiggerGait reuses only the mask branch from BigGait [3], where employing the human foreground is a common practice in gait recognition [3, 4]. We therefore regard this as light gait priors, as illustrated in Fig. 1 (c). As shown in Fig. 2, the central innovation of BiggerGait lies in treating intermediate layers independently to fully unlock the power of large vision models. Furthermore, BiggerGait* (Fig. 6 (b)) introduces a straightforward layer grouping strategy that strikes a balance between performance gains and computational efficiency.
>
> Moreover, we deliberately term our method BiggerGait for two reasons:
> - **(a)** to highlight that it achieves greater (bigger) performance improvements while relying on fewer priors.
> - **(b)** to acknowledge and pay respect to BigGait, the first LVM-based gait recognition method.
>
> Many thanks for your comments, we will improve Fig.1 and Sec. 3.1 for better clarification.
>
> [3] Ye, Dingqiang, et al. "Biggait: Learning gait representation you want by large vision models." Proceedings of the Computer Vision and Pattern Recognition Conference. 2024.
>
> [4] Jin, Dongyang, et al. "On Denoising Walking Videos for Gait Recognition." Proceedings of the Computer Vision and Pattern Recognition Conference. 2025.
>
> ---
>
> ### **\#W3/Q2: Strengthen Justification for Grouping Strategy Design.**
>
> As discussed in Lines 182–189, the grouping strategy of BiggerGait* is designed as a straightforward yet effective approach to achieve a balanced trade-off between accuracy and efficiency. Importantly, while middle layers are indeed complementary (Sec. 3.5), our cosine-similarity-based grouping does not aim to collapse their complementarity. Instead, it clusters the most correlated representations into groups, allowing the network to still capture diversity across groups while avoiding redundant computation within each group.
>
> As shown in Tab. 2 (a) and (b), BiggerGait* (J=6, P=2) achieves accuracy very close to the full version (J=12, P=12) — with only marginal differences (-0.8% CL, -0.2% UP, +0.3% DN, -0.3% BG on CCPG for DINOv2-S-based cases) — while reducing the parameter size by 98.6M and computation by 15.9 GFLOPs. This demonstrates that the proposed grouping strategy is not only simple and reproducible but also practically effective for scaling large vision models in gait recognition.
>
> We agree with the reviewer that there may exist more sophisticated grouping strategies — for example, explicitly maximizing representational dissimilarity or employing learnable grouping as you mentioned. These are promising future directions worth specific exploration. Nevertheless, our current design prioritizes feasibility, reproducibility, and efficiency, which aligns with the main goal of maintaining strong accuracy while significantly reducing computational cost.
>
> Thanks for your suggestion. We will revise the manuscript to explicitly discuss these considerations in the grouping strategy section.
>
> ---
>
> ### **\#W4/Q3: Inference Time.**
>
> The inference time is measured on a single 24GB RTX 6000 GPU using float32 precision, with input resolution of 448×224. We conducted 50 inference runs per model—using the first 30 as warm-up and the remaining 20 to compute the average inference time. The results are as follows:
>
> - Without grouping, DINOv2-S-based BiggerGait requires 53.39 ms / image—significantly slower than BigGait[3] (21.64 ms). However, with the grouping strategy, BiggerGait* achieves a 44% speedup (29.89 ms), dramatically reducing this gap.
> - Silhouette-based methods (GaitBase[5] / DeepGaitV2[6]) spend 13.77 ms on segmentation (DeepLabV3[7]) preprocessing alone, with total times of 21.35 / 22.09 ms, respectively.
>
> These results demonstrate that our grouping strategy is pivotal in bridging the efficiency gap with SoTA approaches, and we will include these data in the manuscript.
>
> [5] Fan, Chao, et al. "Opengait: Revisiting gait recognition towards better practicality." Proceedings of the IEEE/CVF conference on computer vision and pattern recognition. 2023.
>
> [6] Fan, Chao, et al., "OpenGait: A Comprehensive Benchmark Study for Gait Recognition Towards Better Practicality," in IEEE Transactions on Pattern Analysis and Machine Intelligence, doi: 10.1109/TPAMI.2025.3576283.
>
> [7] Chen, Liang-Chieh, et al. "Rethinking atrous convolution for semantic image segmentation." arXiv preprint arXiv:1706.05587 (2017).

---

> ### Author Response · Authors · 2025-08-04
>
> **Hi Reviewer r6GY!**
>
> We sincerely appreciate your huge efforts and kind patience for both us and gait community family.
>
> Your insightful comments have truly helped improve the quality of our manuscript.
>
> We are deeply happy to continue the discussion, if you have any follow-up questions or ideas.

---

> > ### Comment · Reviewer_r6GY · 2025-08-06
> >
> > Thanks for your nice response!
> > For W2-4: My previous concerns are well addressed, and hope the authors would revise the paper accordingly.
> > For W1: I acknowledge the contributions and recognize the efforts to gait recognition field, while I still think the novelty of layer-wise analysis of adopted LVM is somewhat limited. That is, the previous concern is the main factor for changing the rating.
> > In summary, I appreciate the overall contribution to gait recognition field and would raise the rating to 4.

---

> > > ### Author Response · Authors · 2025-08-06
> > >
> > > We’re so happy to hear that most concerns (W2–4) have been well resolved. Your constructive feedback and the improved rating are deeply appreciated. We fully understand your thoughtful concerns regarding W1. The paper will be carefully revised based on these valuable reviews.

---

### Note · Authors · 2025-08-13

Dear all,

We sincerely appreciate the reviewers’ kind patience and the huge effort. Your insightful comments have deeply contributed to improving our manuscript. **Most of all, we feel so happy that the most concerns have been largely resolved during the discussion session.**

- **Reviewer jAUv** strongly supports this work, giving it a highly positive initial rating.

- **Reviewer zEBF** expressed satisfaction with our response and confirmed ongoing support.

- **Reviewer AzAN** stated that our detailed response addressed most of their concerns, and learn towards accepting this paper.

- **Reviewer r6GY** appreciate our overall contribution to gait recognition field and would raise the rating to 4.

**We are truly grateful for all these warm support!** We firmly believe that BiggerGait, together with the reviewers’ valuable comments, can also provide inspiration to employ the diverse intermediate representation from LVMs for other fine-grained vision tasks.

Sincerely,

BiggerGait Authors

---

### Decision · Program_Chairs · 2025-09-17

**Decision:**

Accept (poster)

**Comment:**

The paper introduces BiggerGait, which is a simple baseline for LVM-based gait recognition that leverages complementary representations from intermediate layers of large vision models rather than relying heavily on gait-specific priors. The paper has gone through various discussions between the reviewers and authors in the rebuttal phase, and has eventually received the scores of BA, A, BA, BA. All reviewers acknowledge the contributions of the work to the gait community (e.g., systematic layer-wise analysis), while the key weaknesses are the lack of significant novelty at the core of the method, dependence on dataset/task complexity, and limited temporal modeling.